# Photoactivated Processes on the Surface of Metal Oxides and Gas Sensitivity to Oxygen

**DOI:** 10.3390/s23031055

**Published:** 2023-01-17

**Authors:** Artem Chizhov, Pavel Kutukov, Artyom Astafiev, Marina Rumyantseva

**Affiliations:** 1Chemistry Department, Moscow State University, 119991 Moscow, Russia; 2N.N. Semenov Federal Research Center for Chemical Physics of Russian Academy of Sciences, 119991 Moscow, Russia

**Keywords:** gas sensor, photoactivation, mass spectrometry, nanomaterials, metal oxides

## Abstract

Photoactivation by UV and visible radiation is a promising approach for the development of semiconductor gas sensors with reduced power consumption, high sensitivity, and stability. Although many hopeful results were achieved in this direction, the theoretical basis for the processes responsible for the photoactivated gas sensitivity still needs to be clarified. In this work, we investigated the mechanisms of UV-activated processes on the surface of nanocrystalline ZnO, In_2_O_3_, and SnO_2_ by in situ mass spectrometry and compared the obtained results with the gas sensitivity to oxygen in the dark and at UV irradiation. The results revealed a correlation between the photoactivated oxygen isotopic exchange activity and UV-activated oxygen gas sensitivity of the studied metal oxides. To interpret the data obtained, a model was proposed based on the idea of the generation of additional oxygen vacancies under UV irradiation due to the interaction with photoexcited holes.

## 1. Introduction

Semiconductor gas sensors, which operate on the principle of a reversible change in the electrical resistance in the presence of a detectable gas, are the most important promising sensors for monitoring the composition of the atmosphere, detecting toxic gases and explosive vapors in everyday life and in chemical production, in medical applications, and in many other situations [1,2,3,4]. The advantages of semiconductor gas sensors usually include the simplicity of their design, high sensitivity, miniaturization, and low cost. The disadvantages of semiconductor sensors can include the high power consumption associated with the need to heat the sensitive metal oxide layer to a temperature of 150–500 °C and the drift of the sensor parameters due to the influence of diffusion and recrystallization processes during operation at high temperatures [5,6]. A promising approach to lowering the operating temperature of semiconductor sensors, which was developed significantly in the last decade, is using UV or visible radiation for the activation of and enhancing the gas sensitivity [7,8,9].

Understanding the mechanisms that underlie the gas sensitivity of metal oxides is essential for the targeted development of sensor materials with desired properties [10]. To describe the mechanism of the gas sensitivity of metal oxide sensors, two main models are known. According to the first, (1) the electrical conductivity of the sensing layer is controlled by chemisorbed oxygen species ($O_{2}^{-}$, $O_{2}^{2-}$, O^−^, O^2−^, and others) on the surface of metal oxide grains [11,12], and to the second, (2) the electrical conductivity of the sensing layer is controlled by charged oxygen vacancies ($V_{O}^{+}$ and $V_{O}^{2+}$) near the surface of metal oxide grains [13,14]. Thus, in both models, the interaction of the surface of metal oxides with oxygen occupies a central place in explaining the occurrence of a sensor response to various gases. Surface redox reactions to which involved the molecules of detected gases leads to a change in the concentration of chemisorbed oxygen (oxygen vacancies), whereupon the electrical properties of the sensor layer are changed.

For “classical” semiconductor gas sensors, the reactions of detecting gas molecules on the metal oxide surface occurs when additional kinetic energy is imparted to the interacting particles with an increasing temperature, i.e., through a thermal activation mechanism. Under the light activation, a completely different mechanism takes place, at the initial stage of which nonequilibrium charge carriers are generated, which can then enter into chemical or physicochemical processes with adsorbate molecules, acting as an oxidizing agent (photoexcited holes) or a reducing agent (photoexcited electrons). The capture of photoexcited charge carriers as a result of adsorption or a chemical reaction changes their concentration, and hence photoconductivity, which can be interpreted as a photoactivated sensor signal. The photostimulated reaction of oxygen with the surface of metal oxides, leading to its photoadsorption or photodesorption, is also an important process affecting the photoconductivity of the sensor layer.

The most commonly used materials for gas sensors are ZnO, SnO_2_, In_2_O_3_, and some other metal oxides. All of these metal oxides demonstrate high photoconductivity when exposed to UV radiation and are often considered as materials for photoactivated gas sensors, both individually and as a component of gas-sensitive nanocomposites (see, for example, the references in [15]). In particular, the mentioned metal oxides are also considered as oxygen gas sensors [16,17,18,19,20,21,22], and several papers were recently published on enhancing the sensor response to oxygen under UV activation [23,24,25,26]. Oxygen detection by metal oxide gas sensors, in addition to the obvious practical application, is the simplest case for modeling various mechanisms of gas sensitivity due to the chemical simplicity of such a two-component system (metal–oxygen).

At the same time, the features of the photoactivated reactions of adsorbed gas molecules on the surface of metal oxides, in particular oxygen, have not yet been systematically studied. Among the mentioned metal oxides, ZnO occupies a peculiar position because the study of its photoactivated interaction with oxygen has a long history [27]. To study this interaction, direct methods were used, including manometry [28], mass spectrometry (MS) [29,30], as well as various indirect methods, such as measuring the photoconductivity [31,32,33], contact potential difference [34], and photoluminescent (PL) [35] and electron paramagnetic resonance (EPR) spectroscopy [36]. It was found, in general terms, that depending on the pretreatment of the ZnO samples, both photoadsorption and photodesorption can take place [37]; the photoactivated oxygen isotopic exchange (POIE) was observed over the ZnO and the kinetic parameters of these processes were determined [29]. The maximum quantum yield of the oxygen photoadsorption on the ZnO determined in a similar way was 2.85 eV [38].

On the contrary, direct information about photostimulated oxygen-related reactions on the surface of SnO_2_ and In_2_O_3_ is still insufficient in surface physicochemistry. The photoadsorption of oxygen on the SnO_2_ surface was studied by mass spectrometry [39]; the maximum quantum yield of the photoadsorption was observed at a photon energy of 2.5 eV. Some indirect data are available, such as experiments on photoconductivity and an EPR study of oxygen photoadsorption on SnO_2_ [40]. The POIE over the In_2_O_3_ was studied for samples annealed at different temperatures, and the sample annealed at 200 °C had the highest activity [41].

The aim of this work is to examine the reactivity of ZnO, SnO_2_, and In_2_O_3_ to oxygen by in situ mass spectrometry and compare the obtained results with the gas sensitivity of metal oxides to oxygen by the in situ measurement of the electrical resistance in the dark and under UV activation in the temperature range 30–150 °C. The correlation between the POIE activity and an enhanced UV-activated response to oxygen was found for the studied metal oxides. The experimental data are discussed in terms of the “ionosorption” and “surface conduction” models interlinking the oxygen-related photoactivated processes on the surface of metal oxides with gas-sensitive properties to oxygen.

## 2. Materials and Methods

### 2.1. Synthesis on Nanocrystalline Oxides

(1) Nanocrystalline ZnO was synthesized by decomposition of basic zinc carbonate (#96466, Sigma-Aldrich, St. Louis, MO, USA) in air at 300 °C for 24 h according to the following reaction:(1)$${\mathrm{Zn}}_{5}({\mathrm{CO}}_{3}{)}_{2}(\mathrm{OH}{)}_{6}\to 5ZnO+2{\mathrm{CO}}_{2}+3H_{2}O$$

(2) Nanocrystalline SnO_2_ was synthesized at 70 °C by slowly adding concentrated ammonia solution to a SnCl_4_·5H_2_O (#244678, Sigma-Aldrich, St. Louis, MO, USA) solution in ethanol with stirring until neutral pH was achieved. The reaction resulted in the formation of α-stannic acid gel:(2)$$SnCl_{4}\cdot 5H_{2}O+4{\mathrm{NH}}_{3}+2H_{2}O\to {\mathrm{SnO}}_{2}\cdot nH_{2}O↓+4NH_{4}Cl$$

Resulting gel was separated by centrifugation, then the precipitate was redispersed in 3%-NH_4_NO_3_ solution and again sedimented by centrifugation to remove the chloride ions. More than 12 washing cycles were performed until a negative test for chloride ions with AgNO_3_ solution was achieved. Finally, the SnO_2_·nH_2_O precipitate was washed with ethanol, dried at 70 °C, and annealed at 500 °C in air for 24 h.
(3)$$SnO_{2}\cdot nH_{2}O\to SnO_{2}+nH_{2}O$$

(3) Nanocrystalline In_2_O_3_ was synthesized at 70 °C by slowly adding concentrated ammonia solution to an In(NO_3_)_3_ hydrate (#326135, Sigma-Aldrich, St. Louis, MO, USA) solution in ethanol with stirring until neutral pH was achieved.
(4)$$In(NO_{3}{)}_{3}+3NH_{3}+3H_{2}O\to In(OH{)}_{3}↓+3NH_{4}NO_{3}$$

The precipitate was separated by centrifugation, washed with distilled water and alcohol, dried at 70 °C, and annealed at 500 °C in air to form nanocrystalline In_2_O_3_:(5)$$2In(OH{)}_{3}\to In_{2}O_{3}+3H_{2}O$$

### 2.2. Characterization

Phase composition and crystal structure of synthesized metal oxides was studied by powder X-ray diffraction (XRD) with a Rigaku Max-2500 diffractometer using CuKα radiation (wavelength λ = 1.54059 Å). Average crystallite size *D* was calculated using the Sherrer equation:(6)$$D=\frac{k\lambda }{\sqrt{\beta_{exp}^{2}-\beta_{app}^{2}}\cos\theta }$$
where λ is a wavelength of X-ray radiation, nm; $\beta_{exp}$ is the observed peak width at half height and $\beta_{app}$ is the instrumental broadening, rad; θ is a diffraction angle; *k* is a coefficient equal to 0.89 [42].

The specific surface area of nanocrystalline ZnO was measured by low-temperature nitrogen adsorption using the BET model with the Chemisorb 2750 instrument (Micromeritics Instrument Corporation, Norcross, GA, USA).

Raman spectra were registered with a SENTERRA Raman microscope-spectrometer (Bruker, Billerica, MA, USA) using a 50 × 0.75 NA microscope objective lens and laser excitation at 785 nm.

### 2.3. Gas-Sensing Measurements

The method for examining the gas-sensitive properties of nanocrystalline oxides and the setup for sensor measurements are described in detail in our previous work [43]. Briefly, the gas-sensitive properties of the synthesized oxides were studied by in situ measurement of the electrical resistance of thick (about 100 μm) conducting polycrystalline porous films. Alumina plates 1.5 × 1.5 mm in size with deposited Pt electrodes with a distance of 200 μm between them were used for measurements. On the reverse side of the plates, a platinum spiral was also deposited to heat the sensors with electric current. A thick conductive layer was formed on the measuring plates over the Pt electrodes by annealing at 300 °C a pasty mixture of metal oxide with α-terpineol. The obtained sensors have a linear current–voltage characteristic at bias up to 4 V [43].

To study the gas sensitivity to oxygen, high-purity nitrogen (99.999%) was used as a carrier gas, which was diluted with high-purity oxygen (99.99%) in the concentration range of 0.2–3.2 vol.%. The measurements were carried out in a flow cell at a flow rate of 100 mL/min. A UV LED ($\lambda_{max}=365$ nm) was used for irradiation; the irradiance of the sensors during measurements was 6 $\mathrm{mW}/{\mathrm{cm}}^{2}$. The resistance measurement was carried out on a laboratory-made setup at a bias voltage of 4 V. The sensor signal *S* both in the dark and under continuous UV irradiation was calculated as the ratio of the increase in sensor resistance in the presence of oxygen to its base resistance in dry nitrogen (background environment):(7)$$S=\frac{R_{O_{2}}-R_{N_{2}}}{R_{N_{2}}}$$

### 2.4. In Situ Ambient Pressure UV-Assisted Mass Spectrometry

The setup for in situ mass spectrometric measurements under dark conditions and under UV radiation is described in detail in our previous work [43]. Generally, it consists of a flow PTFE cell (internal volume 15 mL) with a quartz window for irradiation of samples, a set of UV LEDs for irradiation ($\lambda_{max}=365$ nm), an electric heater allows to heat samples up to 150 °C, gas and electrical circuits for controlling the cell. The outlet of the cell is connected to the capillary of the mass spectrometer (MS7–200, equipped with RGA–200 analyzer, Stanford Research Systems). During measurements, a carrier gas is passed through the cell, and the composition of the gas phase at the output of the cell is continuously monitored. Mass spectra were recorded in the range of 4–50 a.m.u. (one scan every 9 s) using an electron multiplier with a gain of 500–1000. All measurements were carried out at ambient pressure.

High-purity helium with an oxygen content of about 15 ppm was used as a carrier gas for oxygen photoadsorption experiments and the typical carrier gas flow rate through the cell was 15 mL/min. For POIE experiments, oxygen-18-enriched water (≥98%, “Center of Molecular Research”, Moscow, Russia) was used. An electrolyte based on oxygen-18-enriched water (saturated Na_2_SO_4_ solution) was placed in an electrolyzer with platinum electrodes. During electrolysis, the current was set to 3 mA, and a carrier gas (helium with 15 ppm of naturally occurring oxygen-16) was passed through the anode space of the electrolyzer, then the carrier gas entered the drying column filled with 3A zeolites and, finally, into the measuring cell with samples. The concentrations of ^16^O_2_ and ^18^O_2_ in the resulting carrier gas were about 15 and 5 ppm, respectively.

For MS studies, the synthesized ZnO, SnO_2_, In_2_O_3_ powders were pressed into tablets with a diameter of 20 mm and a thickness of 1 mm at a pressure of 60 $\mathrm{kg}/{\mathrm{cm}}^{2}$ and 4 tablets were usually placed in the cell; thus, the irradiated surface area of the samples was about 12.5 ${\mathrm{cm}}^{2}$, irradiance of samples at the experiments was 40–60 $\mathrm{mW}/{\mathrm{cm}}^{2}$. Irradiance in MS and gas-sensing experiments was measured by Nova II radiometer equipped with photodiode PD300-UV-193 head (Ophir, Jerusalem, Israel).

## 3. Results and Discussion

### 3.1. Structure and Morphology of Synthesized Metal Oxides

Figure 1a–c show the diffraction patterns of the synthesized powders of the metal oxides. As a result of annealing at 300 °C, the ZnO phase is formed with a wurtzite structure (space group $P6_{3}mc$, 36-1451 card, PDF2) and an average crystallite size of 15 nm, estimated from the broadening (101) reflection using the Scherrer equation. The specific surface area of the synthesized ZnO is 4 $m^{2}/g$. The annealing of the alpha-stannic acid at 500 °C leads to a single-phase SnO_2_ with a cassiterite structure (space group $P4_{2}/mnm$, 41-1445 card, PDF2) [44]; the average crystallite size estimated from the broadening of the (110) reflection is 11 nm, and the specific surface area is 25 $m^{2}/g$. Upon the annealing of the indium hydroxide at 500 °C, the In_2_O_3_ phase with the bixbyite structure is formed (space group $Ia\bar{3}$, 6-416 card, PDF2), the average crystallite size estimated from the broadening of the (222) reflection is 25 nm, and the specific surface area is 10 $m^{2}/g$.

Figure 1d–f show the Raman spectra of the synthesized metal oxides powders in the range of 100–900 ${\mathrm{cm}}^{-1}$. The observed Raman modes confirm the phase composition of the synthesized samples. For ZnO, the most intense $E_{2}\left(\mathrm{high}\right)$ mode is observed at 438 ${\mathrm{cm}}^{-1}$, which corresponds to the vibrations of the oxygen sublattice [45]. Less intense modes $E_{2}\left(\mathrm{high}\right)-E_{2}\left(\mathrm{low}\right)$ and $2E_{2}\left(\mathrm{low}\right)$ at 330 and 204 ${\mathrm{cm}}^{-1}$, respectively, are also observed in the spectrum. In_2_O_3_ demonstrates the most intense modes at 306 and 130 ${\mathrm{cm}}^{-1}$, which correspond to vibrations of the In-O bonds in the InO_6_ octahedrons and the bending vibrations of the InO_6_ octahedrons. The other two peaks at 494 and 628 ${\mathrm{cm}}^{-1}$ refer to the stretching vibrations in the InO_6_ octahedrons, while the mode at 364 ${\mathrm{cm}}^{-1}$ refers to the stretching vibrations of the In-O-In and can also depend on the presence of oxygen vacancies in the structure [46,47]. The spectrum of SnO_2_ contains the main $A_{1g}$ mode at 627 ${\mathrm{cm}}^{-1}$ and a broad Raman peak located around 561 ${\mathrm{cm}}^{-1}$. The high ratio between the peak intensities at 561 and 627 ${\mathrm{cm}}^{-1}$ indicates a high concentration of surface oxygen vacancies on the surface of the SnO_2_. Moreover, a peak with a lower intensity is observed in the spectrum at 309 and 775 cm ${\mathrm{cm}}^{-1}$, which correspond to the $E_{u}$ and $B_{2g}$ Raman modes [48,49].

### 3.2. Oxygen Photoadsorption and UV-Activated Oxygen Isotopic Exchange

Figure 2 shows the mass spectra for m/z = 32 obtained from the analysis of a helium-oxygen carrier gas passed through a cell with samples of metal oxides at temperatures of 30, 50, 100, and 150 °C. UV irradiation was carried out at each temperature for 30 min. The tablets were annealed in air at 300 °C immediately before the measurements. For all three studied samples under UV light, a decrease in the oxygen concentration in the carrier gas was noted, which indicates a shift in the adsorption equilibrium
(8)$$O_{2}(\mathrm{gas})⇌O_{2}(\mathrm{ads})$$
under UV radiation toward the formation of an adsorbed form of oxygen, i.e., the photoadsorption process.

For ZnO, oxygen photoadsorption is clearly detected at room temperature, and with an increasing temperature, the photoadsorption rate slightly increases. For SnO_2_ and In_2_O_3_, the change in the oxygen concentration in the carrier gas under UV light at room temperature is hardly detectable, but at temperatures above 100 °C, oxygen photoadsorption also becomes clearly observed. The oxygen photoadsorption rate decreases during irradiation, which can be explained by the gradual filling of the adsorption sites on the surface of the metal oxides.

Figure 3a shows the results of the POIE experiments on the ZnO. As in the previous case, the experiments were carried out on the freshly annealed in air tablets at temperatures of 30, 50, 100, and 150 °C and under UV irradiation at each temperature for 30 min. A carrier gas that consisted of He + ^16^O_2_ + ^18^O_2_ was used for the POIE experiments.

Initially (in the dark), the signal by m/z = 34 in the mass spectrum has a background intensity, because the ^16^O are contained in ^18^O-enriched water as an impurity (≤1%) and a small amount of mixed isotope ^16^O^18^O molecules is formed as a result of the electrolysis (similarly, due to the content of the ^17^O impurity in the ^18^O-enriched water, a trace amount of particles with a mass of 35 a.m.u. can be detected, which correspond to the molecules of the ^17^O^18^O isotopic composition). A mixture of the oxygen isotopes ^16^O_2_ and ^18^O_2_ gives the main signals by a mass of 32 and 36 a.m.u., respectively, in a ratio of approximately 1:3 (Figure 3b).

Figure 3a shows that the irradiation of ZnO with UV light at each of the temperatures (30, 50, 100, and 150 °C) is accompanied by an increase in the signal with a mass of 34 a.m.u., as well as a decrease in signals with masses of 32 and 36 a.m.u. Thus, as a result of the irradiation, the concentration of the molecules of a mixed isotopic composition ^16^O^18^O increases and the concentration of molecules ^16^O_2_ and ^18^O_2_ decreases. The results obtained indicate that the breaking of the bonds of adsorbed oxygen molecules following the oxygen photoadsorption and the desorption of oxygen molecules, including of molecules with a mixed isotopic composition ^16^O^18^O, occurs. The dependence of the POIE rate on the temperature for ZnO is clearly non-linear, decreasing upon heating to 50 °C compared to room temperature, increasing upon heating to 100 °C, and decreasing again at 150 °C. Apparently, this behavior is associated with the dependence of the rate of thermally activated oxygen desorption from the surface of ZnO on the temperature.

It was also found that for ZnO after a cycle of measurements with heating to 150 °C and subsequent cooling to 30 °C, the POIE rate significantly increases compared to the POIE on the initial (annealed in air) sample at 30 °C (Figure 3a). Assuming that the carrier gas is mainly an inert gas (99.999% He), it can be argued that reductive annealing occurs, which most likely leads to an increase in the density of the surface oxygen vacancies.

Figure 4 shows the results of a comparative POIE experiment on ZnO, In_2_O_3_, and SnO_2_ carried out under similar conditions and also on samples freshly annealed in air. As can be seen, the studied metal oxides demonstrate the different activities in the POIE experiments. The most intense POIE occurred over the ZnO, which is observed in the temperature range of 30–150 °C, while the maximum rate is observed at a temperature of 100 °C. The POIE of a lower intensity (by 5–10 times in comparison with the ZnO) is observed over the In_2_O_3_, also in the temperature range from room temperature to 150 °C, and the rate is weakly dependent on the temperature. On the contrary, no POIE was observed over the SnO_2_ in this experiment. The effect of an increase in the POIE rate at room temperature after the reductive annealing of the samples at 150 °C was also noted for the In_2_O_3_; however, the POIE was still not detected on the surface of the SnO_2_, even after reductive annealing. It should also be noted that a thermally activated oxygen exchange was not observed in the studied temperature range.

### 3.3. CO_2_ Photodesorption

During the experiments, it was found that under UV irradiation and heating from the surface of metal oxides, the photo- and thermal desorption of carbon dioxide occurs. CO_2_ photodesorption was observed using both carrier gases containing the natural oxygen isotope ^16^O_2_ and a mix of ^16^O_2_/^18^O_2_. In the second case, the mass spectra revealed both natural isotopic C^16^O^16^O (44 a.m.u.) and mixed isotopic carbon dioxide C^16^O^18^O (46 a.m.u.). Fully ^18^O-containing C^18^O^18^O molecules with a mass of 48 a.m.u. were not detected in the spectra, even in trace amounts. For each of the studied metal oxides, individual patterns of CO_2_ photodesorption were observed, which are shown in the Figure 5.

The release of the largest amount of CO_2_ under UV irradiation and heating was noted from the surface of the ZnO. Photodesorption peaks are observed at 30 and 50 °C, but much more intense thermal desorption peaks are observed when heated from 30 to 50 °C and from 50 to 100 °C. Above 100 °C, the effects of the photo- and thermal desorption of the CO_2_ almost disappear. The patterns of the photodesorption of the CO_2_ molecules with different isotopic compositions are similar; however, the intensity of the photodesorption peaks for C^16^O^18^O and C^16^O^16^O correlate approximately as 1:100 at 50 °C.

The CO_2_ photodesorption from the surface of the In_2_O_3_ at room temperature is practically not observed; weak photodesorption is observed starting from a temperature of 50 °C. In this case, when heated from 50 to 100 and from 100 to 150 °C, thermal desorption peaks are observed, which have a much higher intensity than the photodesorption peaks. The patterns of the photodesorption of the CO_2_ molecules of various isotopic compositions from the surface of the In_2_O_3_ are similar, although the intensity of the signal between the naturally and mixed isotope molecules is very low (approximately 1:60 for the photodesorption peak at 100 °C).

SnO_2_, like In_2_O_3_, exhibits weak CO_2_ photodesorption behavior, but photodesorption is observed in this case at room temperature. As the temperature increases, the photodesorption rate slightly increases, and typical thermal desorption peaks are also observed. The patterns of the photodesorption of the CO_2_ molecules with different isotopic compositions from the surface are similar; the intensity of the signal from naturally and mixed isotope molecules is very low (approximately 1:60 for the photodesorption peak at 100 °C).

Thus, among the studied samples, the rate of photodesorption CO_2_ at room temperature generally decreases in the series ZnO > SnO_2_ > In_2_O_3_.

### 3.4. Gas Sensitivity to Oxygen

Figure 6 shows the electrical response of the nanocrystalline ZnO, SnO_2_, and In_2_O_3_ to oxygen in the range of 0.2–3.2 vol.% in dark conditions at temperatures of 30, 50, 100, and 150 °C. The oxygen was supplied to the nitrogen flow by pulses lasting 15 min with a distance between pulses of 30 min with an increasing concentration.

The dark electrical resistance of the sensors in a nitrogen atmosphere is different, about $10^{8}$ Ω for ZnO, $10^{5}$ Ω for SnO_2_, and $10^{3}$ Ω for In_2_O_3_. With the introduction of oxygen at a concentration of up to 3.2 vol% at room temperature, only a slight increase in the resistance is observed for all three samples, which corresponds to a sensor signal of the order of 0.01–0.1. In addition, the kinetics of the response and recovery of the resistance of the sensors at room temperature are very slow and do not reach saturation during the oxygen supply period. An increase in the temperature leads to both an increase in the sensor signal and an acceleration of the kinetics of the response and decay times of the sensors, while good reversibility is observed only starting from a temperature of 150 °C. At the same time, the metal oxides show close values of the sensor signal to oxygen, for example, in dark conditions at a temperature of 100 °C, the sensor signals to 0.8 vol.% oxygen for oxides of ZnO, SnO_2_, and In_2_O_3_ are 0.27, 0.37, and 0.16, respectively. An increase in the temperature to 150 °C leads to an increase in the sensor signal to 2.07, 1.19, and 2.04, but the obtained values are still close to each other.

The effect of UV irradiation on the electrical response of nanocrystalline ZnO, SnO_2_, and In_2_O_3_ to oxygen is shown in Figure 7. It can be seen, firstly, that UV irradiation leads to a significant drop in the baseline resistance of the sensors in a nitrogen atmosphere, to $10^{4}$ Ω for the ZnO and SnO_2_ and up to $10^{2}$ Ω for the In_2_O_3_. Moreover, even at room temperature, there is a significant sensor response to the oxygen and a decrease in the response and recovery times. Simultaneous heating to 150 °C and UV irradiation lead to an increase in the sensor signal of the samples compared to dark conditions at the same temperature. Both in the dark and under UV irradiation, the resistance of ZnO-, SnO_2_-, and In_2_O_3_-based sensors increases in the presence of oxygen relative to the base resistance in nitrogen, which implies a typical response of an *n*-type semiconductor to an oxidizing gas.

UV irradiation leads to different effects on the sensor signal for three other metal oxides. Figure 8a shows the temperature dependence of the sensor signal to 0.8% oxygen in the dark ($S_{\mathrm{dark}}$). It can be seen that the ZnO-, SnO_2_-, and In_2_O_3_-based gas sensors demonstrate a small, close-to-each-other sensor signal in a temperature range of 30–150 °C. With an increasing temperature, the sensor signal increases; in semilogarithmic coordinates, a dependence $S_{\mathrm{dark}}\left(T\right)$ close to linear is observed, the slope of which is approximately the same for all three oxides.

Figure 8b shows the temperature dependence of the sensor signal to 0.8% oxygen under UV irradiation ($S_{\mathrm{UV}}$). It can be seen that the sensor signal for all the studied oxides increases significantly compared to the $S_{\mathrm{dark}}$ and increases with an increasing temperature. For the SnO_2_ and In_2_O_3_, the photoactivated sensor signal begins to increase significantly at temperatures above 100 °C. For them, the $S_{\mathrm{UV}}\left(T\right)$ is close to linear in the semilogarithmic coordinates but with different slopes: for the In_2_O_3_, a steeper slope is observed, and for the SnO_2_, it is flatter. In the case of ZnO, there is an increase in $S_{\mathrm{UV}}$ starting from 50 °C; however, at 150 °C, the sensor signal reaches a plateau, so the dependence is no longer linear. According to the absolute value of $S_{\mathrm{UV}}$, ZnO demonstrates the highest sensor signal toward oxygen compared to SnO_2_ and In_2_O_3_.

Figure 8c shows the temperature dependence of the ratio $S_{\mathrm{UV}}/S_{\mathrm{dark}}$ calculated from the data in Figure 8a,b. It can be seen that for the SnO_2_ and In_2_O_3_, the $S_{\mathrm{UV}}/S_{\mathrm{dark}}$ ratio decreases with an increasing temperature, because the $S_{\mathrm{dark}}$ values increase faster with an increasing temperature than the $S_{\mathrm{UV}}$. The dependence $S_{\mathrm{UV}}/S_{\mathrm{dark}}$ on the temperature for the ZnO has a fundamentally different character, for which a hump arises at temperatures of 50–100 °C, caused by an extremely sharp increase in the sensor signal under UV irradiation (about 40 times). It is also clearly seen in Figure 8b,c that the greatest enhancement of the sensor signal under UV irradiation takes place for ZnO; for In_2_O_3_, high ratios of $S_{\mathrm{UV}}/S_{\mathrm{dark}}$ are also observed; but for SnO_2_, UV irradiation has the least effect on enhancing the sensor signal to oxygen. Figure 8d shows the concentration dependence of the sensor signal to oxygen both in the dark and under UV irradiation at 100 °C. In all the metal oxides, the dependence in the log-log coordinates is close to linear, differing slightly in the slope, i.e., it follows the equation $S\propto C_{O_{2}}^{k}$. For ZnO, *k* = 1 is observed in the dark and under UV; for In_2_O_3_, *k* is also the same in the dark and under UV and equals to 0.8; and for SnO_2_, *k* equals to 0.85 in the dark and 0.95 under UV.

## 4. Discussion

The MS data showed that on the surface of the ZnO, In_2_O_3_, and SnO_2_ freshly annealed in air, the photoadsorption of oxygen is observed under UV irradiation. From the data obtained, it follows that these metal oxides are also characterized by the irreversible photoadsorption of oxygen under UV irradiation, while the photoadsorption rate increases with heating. At the same time, the photoadsorption of oxygen on SnO_2_ and In_2_O_3_ proceeds with a lower rate compared to ZnO. This behavior cannot be explained, for example, by the influence of the specific surface area of the samples, because ZnO, on the contrary, has the smallest specific surface area but the highest photoadsorption rate among the studied samples.

POIE experiments are a continuation of photoadsorption experiments, which allow a more detailed understanding of the mechanism of ongoing processes. The experimental results show that the POIE activity differs significantly for the studied metal oxides and decreases in the ZnO-In_2_O_3_-SnO_2_ series. Because the natural ^16^O isotope is contained both in the composition of the synthesized metal oxides and in the composition of the carrier gas, it is impossible to state whether the POIE occurs between oxygen from the gaseous and solid phases or whether the exchange occurs between oxygen molecules ^16^O_2_ and ^18^O_2_ simultaneously photoadsorbed from the gas phase on the metal oxide surface. However, it can be argued that when the POIE is taking place, photoadsorbed oxygen molecules undergo further breaking of the intermolecular bonds, as a result of which oxygen molecules of a mixed isotopic composition ^16^O^18^O appear in the gas phase. In this case, the appearance of ^16^O^18^O molecules in the carrier gas indicates that the process of the photodesorption of oxygen also occurs during UV irradiation, although, in general, the adsorption equilibrium is shifted toward the formation of a photoadsorbed form of oxygen.

The comparison of the results of the MS and gas-sensing measurements suggests that there is a correlation between the POIE activity of the metal oxide and the photoactivated sensor response to oxygen (Figure 9). ZnO shows the highest POIE activity and UV irradiation also most evidently enhances its sensor response to oxygen; on the contrary, SnO_2_ does not show POIE activity and it is also characterized by the lowest sensor signal to oxygen and has the least effect of UV irradiation on it. In_2_O_3_ occupies an intermediate position, demonstrating slight POIE activity and both a moderate sensor response to oxygen and the mean effect of UV irradiation on it.

An attempt to interpret the results obtained in terms of the known models encounters difficulties. In the case of the “ionosorption model”, it is assumed that oxygen from the gas phase can be adsorbed on the grains’ surface with the capture of a conduction electron:(9)$$O_{2}(\mathrm{ads})+e^{-}⇌{O_{2}}^{-}(\mathrm{ads})$$

Apparently, in the low-temperature region (below 150 °C), chemisorbed oxygen retains the molecular form, while the existence of other charge forms of chemisorbed oxygen remains questionable. By extending the ionosorption model to the case of photoactivation, additional processes are possible with the participation of photogenerated charge carriers. Photogenerated electrons can cause the adsorption of an additional number of oxygen molecules, i.e., photoadsorption discussed above, while photogenerated holes are responsible for the photodesorption of chemisorbed oxygen molecules.
(10)$$O_{2}(\mathrm{ads})+e^{-}(ph)⇌{O_{2}}^{-}(\mathrm{ads})$$
(11)$$O_{2}(\mathrm{ads}{)}^{-}+h^{+}(ph)⇌O_{2}(\mathrm{ads})$$

Thus, the ionosorption model does not provide special ways for the isotopic exchange of chemisorbed oxygen molecules (below 150 °C) and therefore cannot be used to adequately describe the photoactivated processes on the ZnO surface.

On the other hand, a well-developed kinetic model of the photoactivated oxygen exchange on the surface of metal oxides (ZnO, TiO_2_) is known [29,50], according to which the POIE occurs at active centers, which are surface oxygen atoms that have captured a hole, $O_{s}^{2-}$ + h^+^ → $O_{s}^{-}$, which further interacts with gaseous molecules ^18^O_2_ forming a shortliving intermediate three-atomic complex ${O_{3}}^{-}$ during the lifetime of which the heteroexchange occurs:(12)Os−16+O218→O18O18Os−16→O18O16+Os−18

However, such a model developed for photocatalytic materials does not consider the effect of surface processes on the electrical conductivity of metal oxide, i.e., its gas-sensing properties. Moreover, the authors in [50] show that the POIE is much higher on oxidized TiO_2_ than on the reduced sample, while we, in the case of ZnO and In_2_O_3_, are faced with the opposite phenomenon. In addition, the abovementioned model does not take into account the role of surface oxygen vacancies in the reactivity of metal oxides.

At present, many works draw attention to the fact that an increase in the density of oxygen vacancies in ZnO enhances its reactivity, which is manifested in the enhancement of the catalytic, gas sensing, and other properties of V_O_-enriched ZnO [51]. For example, in the work by [52], it was also demonstrated that the photocatalytic activity of ZnO increases significantly when it is thermally treated in a vacuum at 240–260 °C, and the studies also indicated the role of oxygen vacancies in increasing the photocatalytic activity. In a previous article [43], we noted that the reductive annealing of ZnO in He at 150 °C leads to an increase in the oxygen photoadsorption rate. A possible mechanism is an increase in the density of surface oxygen vacancies, which are typical adsorption sites for oxygen molecules, which leads to an increase in the photoadsorption rate. The increase in the concentration of oxygen vacancies in ZnO as a result of reductive annealing was revealed by XPS. In this work, it was found that the reductive annealing of ZnO and In_2_O_3_ in an inert gas at 150 °C also leads to an increase in the POIE rate at room temperature, compared with the POIE rate on freshly annealed in air samples. Thus, the key role of oxygen vacancies in the POIE and enhanced photoactivated oxygen-sensing conditions can be assumed.

An attempt to describe the processes of enhancing the oxygen sensor response under UV activation assisted with simultaneous POIE (which is most evident in the case of ZnO) can be based on a model of surface conductivity, which assumes that the conductivity of metal oxide is controlled by the concentration of surface oxygen vacancies. In the low-temperature region without photoactivation, oxygen chemisorption with the participation of doubly charged oxygen vacancies and conduction electrons can occur with the maintenance of its molecular form with forming the O_2_ − V_O_ intermediate at the first stage [13,53,54]:(13)$$O_{2}\left(\mathrm{gas}\right)+V_{O}^{2+}+2e^{-}⇌O_{2}-V_{O}$$

Varying the concentration of charged oxygen vacancies on the grains’ surface affects the electron density in the electron accumulation layer in the near-surface region, which formed for screening the positive surface charge. Under a change in the partial pressure of oxygen, the equilibrium of the reaction (Equation 13) shifts in the corresponding direction, which provides a reversible sensor response to oxygen in the dark by the changing of the surface conductivity. The maintenance of the molecular form of adsorbed oxygen follows from the fact that under dark conditions up to 150 °C, a sensor signal to oxygen is observed for all metal oxides, but a thermally activated oxygen isotopic exchange is not registered. Under photoactivation, an additional number of electrons are generated that can enter into the above process (Equation 13) and cause the adsorption of an additional number of oxygen molecules. This path is the “conventional” way of photoactivation available for all the considered metal oxides.

Further, in the case of some metal oxides, a significant increase in the sensors’ response can be described by a model based on the formation of an additional number of surface oxygen vacancies during UV irradiation.

For ZnO, the fact is known that UV irradiation can lead to its photolysis, which is a consequence of the oxidation of lattice oxygen by photoexcited holes:(14)$$O^{2-}+2h^{+}\to \frac{1}{2}O_{2}\left(\mathrm{gas}\right)+V_{O}$$

The ionization of extra vacancies generated during UV irradiation proceeds by the reaction with photogenerated holes:(15)$$V_{O}+h^{+}\to V_{O}^{+}$$
(16)$$V_{O}^{+}+h^{+}\to V_{O}^{2+}$$

The formation of $V_{O}^{2+}$ will lead to an increase in the concentration of adsorption sites for O_2_ molecules and a more evident effect on the change in electrical conductivity under UV irradiation and in the presence of O_2_.

Oxygen vacancies on the surface of ZnO can also be considered as centers for the dissociative adsorption of oxygen molecules:(17)$$O_{2}\left(\mathrm{gas}\right)+2V_{O}^{2+}+4e^{-}\to 2O_{O}$$

Thus, photoadsorbing oxygen molecules of different isotopic compositions undergo photodissociation, filling the oxygen vacancies, and then the same oxygen atoms undergo photooxidation by holes according to Equation (Equation 14) with the formation of various isotopes of oxygen atoms, which, combined with each other during desorption, can release oxygen molecules of a mixed isotopic composition ^16^O^18^O.

One can also assume a possible process of photooxidation of the adsorbed oxygen molecule in the O_2_ − V_O_ intermediate, which will also lead to the breaking of bonds of the chemisorbed oxygen molecule and the formation of a vacancy:(18)$$O_{2}-V_{O}+h^{+}\to O_{O}+\frac{1}{2}O\left(\mathrm{gas}\right)+V_{O}^{+}$$

Thus, in the proposed model, photogenerated holes lead to the formation of new surface oxygen vacancies, which can be additionally oxidized to a doubly charged $V_{O}^{2+}$ by the interaction with photoexcited holes (Equations (Equation 15) and (Equation 16)). The generation of an additional number of $V_{O}^{2+}$ during UV irradiation contributes to a greater effect on the electrical conductivity at the adsorption of oxygen molecules, and in addition, they are the centers for the photoactivated oxygen dissociation according to the reactions (Equation 14)–(Equation 18). Oxygen vacancies, previously formed as a result of reductive annealing, immediately provide an increased number of sites for oxygen chemisorption, which leads to an increase in the POIE rate on the reduced samples.

Accordingly, a similar (but less pronounced) mechanism for the generation of additional oxygen vacancies during the irradiation period by means of photogenerated holes can be expected in the case of In_2_O_3_, while for SnO_2_, such a mechanism does not occur, which leads to a poor photoactivated sensor signal to oxygen and the absence of POIE.

The proposed assumptions are confirmed by many literature data on experimental and theoretical studies of processes involving oxygen on the surface of metal oxides. Photolytic ZnO decomposition in a vacuum under UV light was intensively studied by mass spectrometry and manometry in the 1970s–1980s of the XX century (see, for example, the references in [55]), and it was found that about 100 surface monolayers undergo photodecomposition. In a recently published work [56], the authors reported an increase in the defective luminescence of ZnO upon prolonged exposure to UV light, which may also correspond to the formation of additional oxygen vacancies due to photodecomposition. Concepts related to the effect of photodecomposition on surface hydroxylation and structural changes in ZnO sol-gel films were also developed by Asakuma et al. [57,58]. The possibility of oxygen adsorption on ZnO proceeding with the filling of an oxygen vacancy was experimentally demonstrated using in situ EPR and PL spectroscopy [59]. The calculations given in the work [60,61] show that on the reduced surface of ZnO, dissociative oxygen adsorption is preferable to molecular adsorption. In addition, extremely low activation barriers for surface oxygen adsorption and bulk diffusion were estimated to be 0.093 and 0.67 eV, respectively [62].

Indium oxide is a much less explored material than zinc oxide. There are only some references in the literature about the possibility of In_2_O_3_ photodecomposition under UV light [63,64]. Some calculations also confirm the preference for the dissociative adsorption of oxygen on its surface [65]. No information was found in the literature on the possibility of the photolytic decomposition of SnO_2_ under UV irradiation. The calculated data indicate both a dissociative and molecular character of the oxygen adsorption on the surface of SnO_2_ [66,67,68].

It is also interesting to note that the reactivity of the studied metal oxides to oxygen under UV activation correlates with the metal–oxygen bond energy. Thus, for ZnO, the Zn − O bond energy is 10.4 eV; for In_2_O_3_, the In − O bond energy is 12.4 eV; and for SnO_2_, the Sn − O bond energy is 15.5 eV. In general, the lower metal–oxygen bond energy implies an easier loss of oxygen from the crystal lattice and the formation of oxygen vacancies. The given bond energy data fit into the sequence ZnO > In_2_O_3_ > SnO_2_, which also follows for the POIE activity and UV-enhanced sensor response of the studied oxides. The concept of the relationship between the metal–oxygen bond energy and the surface reactivity and the concentration of active centers was considered in detail by Marikutsa et al. [69], and the method for calculating the metal–oxygen bond energy is also given there. Within these ideas, it can be noted that, for example, WO_3_ is characterized by a much higher metal–oxygen bond energy (36.6 eV) and is a suitable example for studying and testing the model put forward in this article.

It is worth mentioning that some articles also discuss the mechanism of a photoassisted oxygen release through the oxidation of carbon impurities and the desorption of CO_2_ [70]. The studies carried out by the method of mass spectrometry showed that, indeed, the photodesorption of CO_2_ from the surface of metal oxides, in particular ZnO, is observed. Taking into account the possibility of the photodesorption of CO_2_ also provides a good explanation for why the photoadsorption of oxygen on the surface of oxides is allegedly irreversible. In fact, one of the channels for the release of adsorbed oxygen is its desorption in the form of CO_2_ molecules. It is interesting to note that for ZnO, the photodesorption of CO_2_ is maximum at temperatures from room temperature to 50 °C, whereas at 100 °C, the CO_2_ photodesorption almost ceases, but the POIE rate, on the contrary, increases, which may indicate that the photooxidation of carbon with the formation of CO_2_ and the photodesorption of oxygen are competing processes. However, unlike oxygen, mixed isotopic molecules C^16^O^18^O are registered in the photo- and thermal desorption patterns of all three oxides, which may indicate different pathways from their origin. There is also no clear correlation between the oxygen-sensing properties of metal oxides and the photo- and thermally stimulated CO_2_ release.

## 5. Conclusions

As a result, in this work, using nanocrystalline ZnO, In_2_O_3_, and SnO_2_, we have shown that there is a correlation between the POIE activity and the photoactivated sensor response to oxygen under UV irradiation (365 nm). According to the data obtained, in the series ZnO–In_2_O_3_–SnO_2_, these characteristics decrease. ZnO shows outstanding POIE activity and an enhanced gas sensitivity to oxygen under UV irradiation, which can be explained by the unique lability of oxygen in its composition. The data obtained are discussed within the framework of a model that assumes the formation of additional oxygen vacancies during UV irradiation through an interaction with photoexcited holes. Additional oxygen vacancies enhance the sensor response to oxygen and act as centers for the photodissociation of chemisorbed oxygen molecules. The proposed model can be supplemented and modified by expanding the range of studied metal oxides and non-oxide materials, including such promising materials as WO_3_, TiO_2_, Ga_2_O_3_, BiOX (X = Cl, Br, and I), and others.

## Figures and Tables

**Figure 1 sensors-23-01055-f001:**
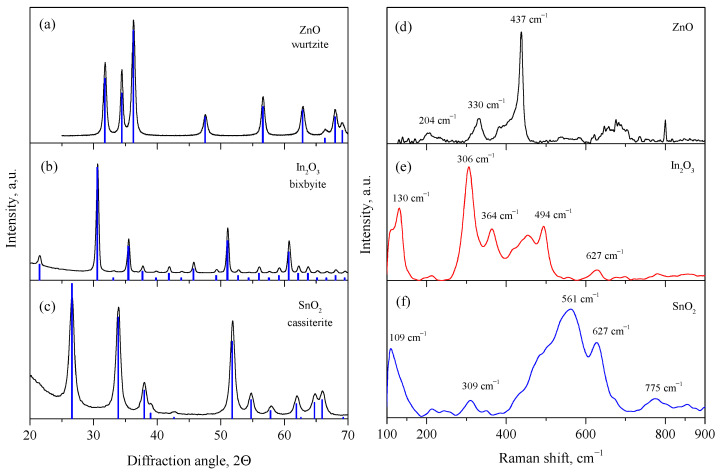
XRD patterns of synthesized nanocrystalline ZnO (**a**), In_2_O_3_ (**b**), and SnO_2_ (**c**) powders and Raman spectra of synthesized nanocrystalline ZnO (**d**), In_2_O_3_ (**e**), and SnO_2_ (**f**) powders.

**Figure 2 sensors-23-01055-f002:**
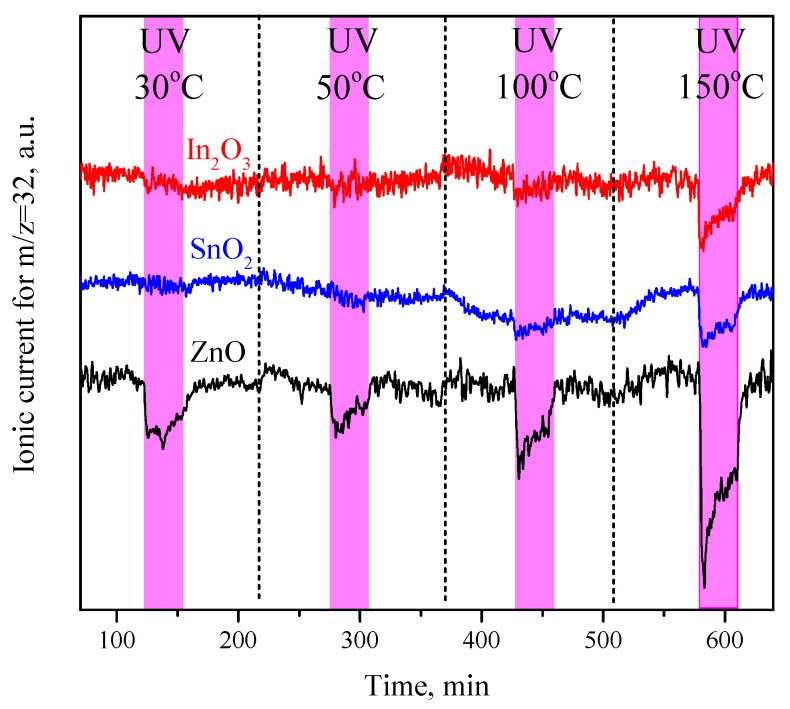
MS analysis of the oxygen content (^16^O_2_, 32 a.m.u.) in the carrier gas passed through the cell with ZnO, SnO_2_, and In_2_O_3_ tablets and the effect of UV irradiation on its concentration at 30, 50, 100, and 150 °C.

**Figure 3 sensors-23-01055-f003:**
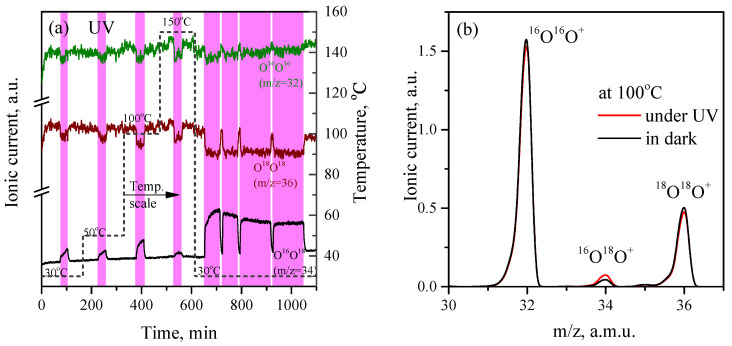
(**a**) MS analysis of the content of oxygen molecules with various isotopic compositions (^16^O^16^O, 32 a.m.u.; ^16^O^18^O, 34 a.m.u.; ^18^O^18^O, 36 a.m.u.) in the carrier gas passed through the cell with ZnO tablets and the effect of UV irradiation on its concentration at temperatures of 30, 50, 100, and 150 °C. After sample cooling, the effect of enhancement of POIE at room temperature is observed. (**b**) Mass spectrum of the carrier gas in the range of 30–37 a.m.u. at 100 °C in dark and under UV irradiation (350th and 400th minutes of the experiment in (**a**)).

**Figure 4 sensors-23-01055-f004:**
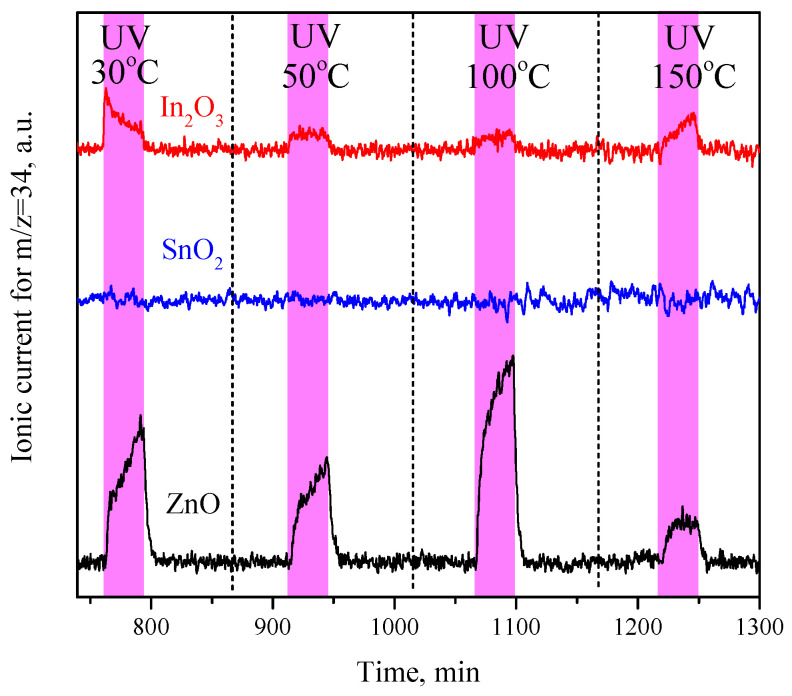
MS analysis of the content of mixed isotopic oxygen (^16^O^18^O, 34 a.m.u.) in the carrier gas passed through the cell with ZnO, SnO_2_, and In_2_O_3_ tablets and the effect of UV irradiation on its concentration at temperatures of 30, 50, 100, and 150 °C.

**Figure 5 sensors-23-01055-f005:**
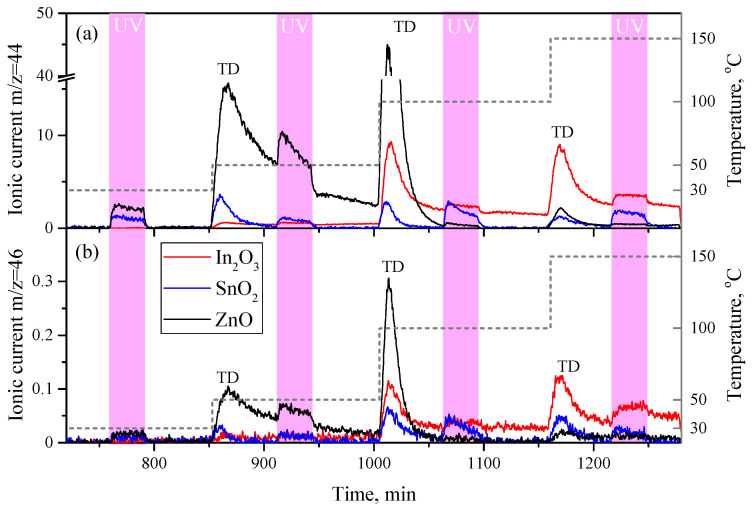
MS analysis of the content of carbon dioxide molecules C^16^O^16^O (44 a.m.u.) (**a**) and C^18^O^16^O (46 a.m.u.) (**b**) in the carrier gas passed through the cell with metal oxide tablets (ZnO, In_2_O_3_, SnO_2_) and the effect of UV irradiation on their concentration at temperatures of 30, 50, 100, and 150 °C. (**a**) is broken on the intensity axis from 16 to 40 a.u. Thermally activated desorption of CO_2_ is marked by “TD” symbols.

**Figure 6 sensors-23-01055-f006:**
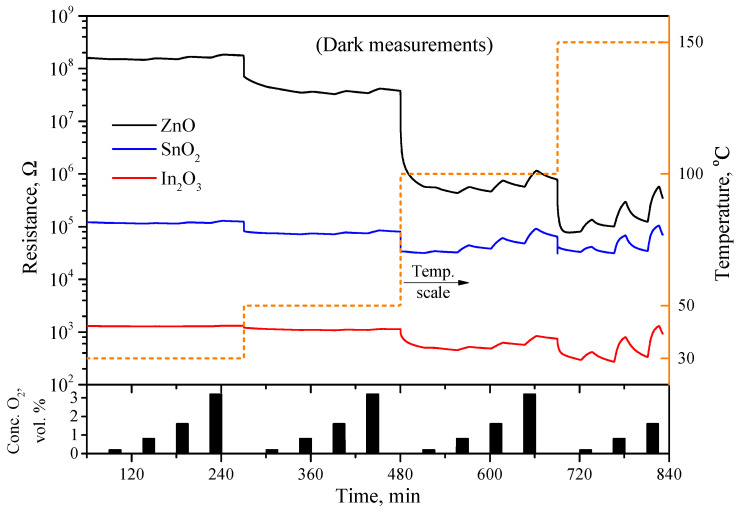
Electrical resistance of ZnO-, SnO_2_-, and In_2_O_3_-based sensors in dark depending on the content of oxygen in gas phase in range of 0.2–3.2 vol.% at temperatures 30, 50, 100, and 150 °C. Nitrogen was used as background gas.

**Figure 7 sensors-23-01055-f007:**
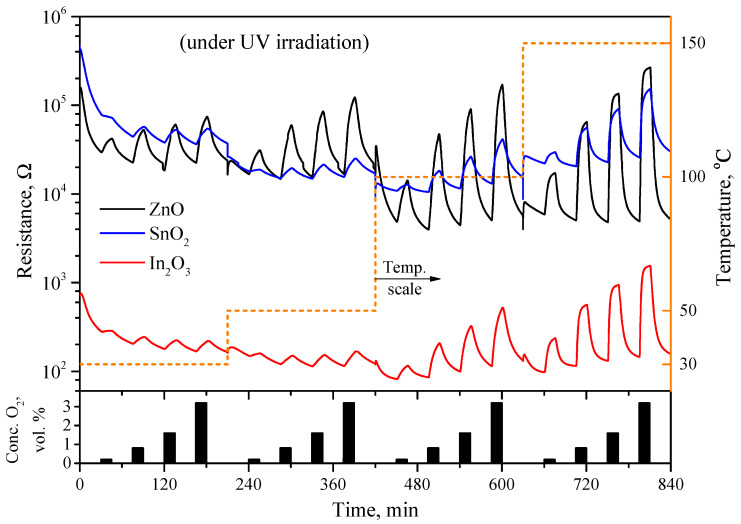
Electrical resistance of ZnO-, SnO_2_-, and In_2_O_3_-based sensors under continuous UV irradiation ($\lambda_{\max}$ = 365 nm) depending on the content of oxygen in gas phase in range of 0.2–3.2 vol.% at temperatures 30, 50, 100, and 150 °C. Nitrogen was used as background gas.

**Figure 8 sensors-23-01055-f008:**
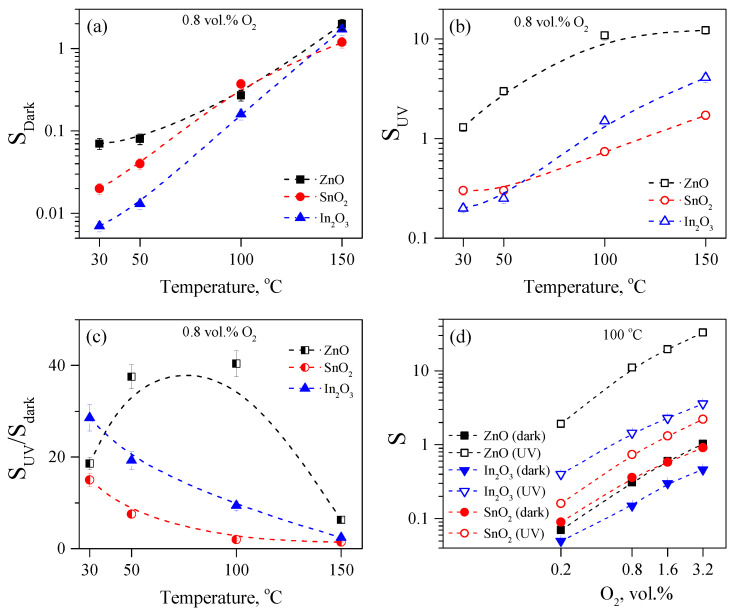
(**a**) Temperature dependence of the sensor signal to 0.8 vol.% oxygen on temperature in the dark ($S_{\mathrm{dark}}$); (**b**) temperature dependence of sensor signal to 0.8 vol% oxygen under UV ($S_{\mathrm{UV}}$); (**c**) temperature dependence of the $S_{\mathrm{UV}}/S_{\mathrm{dark}}$ ratio to 0.8 vol.% oxygen; (**d**) dependence of the sensor signal on the oxygen concentration at 100 °C, both in the dark and under UV irradiation.

**Figure 9 sensors-23-01055-f009:**
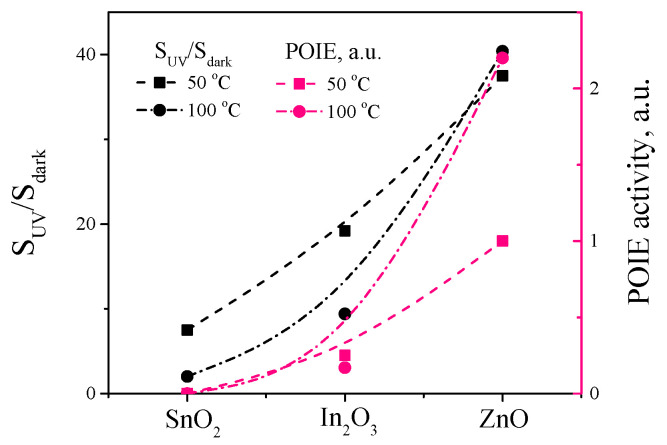
Correlation between $S_{\mathrm{UV}}/S_{\mathrm{dark}}$ and POIE activity for ZnO, In_2_O_3_, and SnO_2_ at 50 °C (squares) and 100 °C (circles).

## Data Availability

The data presented in this study are available upon request from the corresponding author. The data are not publicly available due to privacy reasons.

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
