# Peer review of "Photoactivated Processes on the Surface of Metal Oxides and Gas Sensitivity to Oxygen"

_sensors, 2023, doi:10.3390/s23031055_

Round 1

Reviewer 1 Report

In this paper, the authors investigated the mechanism of UV-activated processes on the surface of ZnO, In2O3, and SnO2 by in situ mass spectrometry and proposed a model based on the idea of generation of additional oxygen vacancies under UV irradiation due to interaction with photoexcited holes. It provides a theoretical basis for explaining the mechanism of photoactivated gas sensitivity. However, there are several issues to be addressed before this manuscript can be considered for publication (Major revision).

1.     Please confirm the formula (6) again and note relevant references.

2.     In the results and discussion section, the analysis of XRD in Figure 1(a-c) should add relevant references, such as Sensors and Actuators: B. Chemical, 326 (2021) 128819.

3. Figure 4 shows the results of a comparative POIE experiment on ZnO, In2O3 and SnO2. Please explain the reason why the POIE intensity of ZnO is higher than that of  In2O3 and SnO2 in the temperature range of 30-150 ℃.

4. Figure 7 shows the electrical resistance of ZnO-, SnO2- and In2O3-based sensors under continuous UV irradiation at temperatures 30, 50, 100 and 150 ℃. Generally, the resistance in air of metal oxides decreased with the operating temperature increasing. However, the resistance of SnO2 at 150 ℃ is higher than that at 100 ℃ in Figure 7. Does the UV irradiation also have the effect on the resistance of metal oxides?Please give a explain.

5. Please add error bar in Figure 8.

6.Why the ZnO sample has the smallest specific surface area but the highest photoadsorption rate among the studied samples? Dose the photoadsorption rate is related to other factors?

7.the authors in [48] shows that the POIE are much higher on oxidized TiO2 than on the reduced sample, while we, in the case of ZnO and In2O3, are faced with the opposite phenomenon. Please analyze the reasons why the results are opposite.

8. The authors said that the key role of oxygen vacancies in POIE and enhanced photoactivated oxygen sensing conditions can be assumed. Please provide evidence of XPS or EPR to characterize the oxygen vacancy of the samples, rather than just assumed the conclusion.

Author Response

In this paper, the authors investigated the mechanism of UV-activated processes on the surface of ZnO, In2O3, and SnO2 by in situ mass spectrometry and proposed a model based on the idea of generation of additional oxygen vacancies under UV irradiation due to interaction with photoexcited holes. It provides a theoretical basis for explaining the mechanism of photoactivated gas sensitivity. However, there are several issues to be addressed before this manuscript can be considered for publication (Major revision).

Dear reviewer, thank you for positive evaluating our work, we will do our best to clarify and correct all shortcomings.

  1. Please confirm the formula (6) again and note relevant references.

Answer: Formula (6) is generally correct. The broadening of the X-ray reflection is calculated from the observed broadening and the instrumental broadening. The relevant link has been added. The capital letter 'theta' in the formula has been corrected to lowercase 'theta'. The coefficient k has been estimated to 0.89 instead 0.9.

  1. In the results and discussion section, the analysis of XRD in Figure 1(a-c) should add relevant references, such as Sensors and Actuators: B. Chemical, 326 (2021) 128819.

Answer: the corresponding reference was added when discussing the results (line 172)

  1. Figure 4 shows the results of a comparative POIE experiment on ZnO, In2O3 and SnO2. Please explain the reason why the POIE intensity of ZnO is higher than that of In2O3 and SnO2 in the temperature range of 30-150 ℃.

Answer: Thank you for your comment. In our opinion, the increase in the rate of oxygen exchange occurs according to the photolytic-dissociative mechanism, which is discussed on lines 434-463. Chemisorbed oxygen molecules on the surface of ZnO undergo dissociative adsorption, filling oxygen vacancies. At the same time, near-surface oxygen atoms in ZnO lattice can be oxidized to O(g) atoms by photoexcited holes. The combination of O atoms of different isotopic composition will lead to the appearance of oxygen molecules of mixed isotopic composition, which is noted in the figure 4. A key feature of ZnO is its ability to photolyze with the formation of oxygen vacancies, while other oxides do not.

  1. Figure 7 shows the electrical resistance of ZnO-, SnO2- and In2O3-based sensors under continuous UV irradiation at temperatures 30, 50, 100 and 150 ℃. Generally, the resistance in air of metal oxides decreased with the operating temperature increasing. However, the resistance of SnO2 at 150 ℃ is higher than that at 100 ℃ in Figure 7. Does the UV irradiation also have the effect on the resistance of metal oxides?Please give a explain.

Answer: You are absolutely right that the resistance of metal oxides should decrease with increasing temperature, because semiconductors are characterized by a positive temperature coefficient of conductivity. However, for the considered metal oxides, a deviation from this law can be observed. As the temperature increases, the reactivity of the metal oxide surface also increases. Chemisorbed oxygen molecules can capture thermally and photoexcited electrons (O2 + e- = O2-), which leads to a decrease in the electron concentration in metal oxide grains and a decrease in electrical conductivity.

  1. Please add error bar in Figure 8.

Answer: Error bars have been added to the figure 8.

6. Why the ZnO sample has the smallest specific surface area but the highest photoadsorption rate among the studied samples? Dose the photoadsorption rate is related to other factors?

Answer: Because our research shows that the nature of the oxide is more important than its characteristics, such as specific surface area, etc. In addition, photoactivated processes occur only on the irradiated surface, which is determined primarily by the geometric parameters of the sample and the roughness of the surface and does not depend on the specific surface area.

7.“the authors in [48] shows that the POIE are much higher on oxidized TiO2 than on the reduced sample, while we, in the case of ZnO and In2O3, are faced with the opposite phenomenon.” Please analyze the reasons why the results are opposite.

Answer: Thank you for your comment. The reasons for this are analyzed in detail in the text below (lines 413-459); in our opinion, in the case of ZnO and In2O3, it is oxygen vacancies that act as centers of dissociative adsorption of oxygen molecules and further photooxidation of surface oxygen atoms by photoexcited holes, which ultimately leads to the appearance in the gaseous phase of mixed-isotopic oxygen molecules 16O18O, i.e. photoinduced oxygen isotopic exchange.

  1. The authors said that the key role of oxygen vacancies in POIE and enhanced photoactivated oxygen sensing conditions can be assumed. Please provide evidence of XPS or EPR to characterize the oxygen vacancy of the samples, rather than just assumed the conclusion.

Answer: XPS characterization of oxygen vacancies in the same ZnO was presented in our previous work (Chizhov, A. et. al. UV-Activated NO2 Gas Sensing by Nanocrystalline ZnO: Mechanistic Insights from Mass Spectrometry Investigations. Chemosensors 2022, 10, 147. https://doi.org/10.3390/chemosensors10040147). There, we actually showed that the concentration of oxygen vacancies in ZnO increases during reduction annealing (we give a figure below from this article; O1s(II) state refers to O atoms near oxygen vacancies).

The following phrase with corresponding reference has been added to the text:

“The increase in the concentration of oxygen vacancies in ZnO as a result of reductive annealing was revealed by XPS.”

Reviewer 2 Report

Although the authors have discussed several weaknesses of this paper, like the relationship between O2 sensing and CO2 release. The stability of O2 sensing and the change of POIE with time are required.

Author Response

Although the authors have discussed several weaknesses of this paper, like the relationship between O2 sensing and CO2 release. The stability of O2 sensing and the change of POIE with time are required.

Answer: Thank you for your comment. The purpose of this work was to investigate the sensing mechanism, but not to present a high performance oxygen sensor, so we did not monitor the long term stability of materials for oxygen detection. Moreover, we do not claim that the sensors presented in this paper have record-breaking characteristics in sensitivity, stability, etc., we only tried to understand what photoactivated processes can be responsible for the oxygen gas sensitivity of ZnO, SnO2, In2O3. We have considered in detail the mechanisms responsible for the photoactivated detection of oxygen, but the discussion of possible ways of degradation of the sensor properties of metal oxides is, although very important, a separate issue that should be devoted to special studies. The stability of POIE is also an interesting issue, however, at this stage of the work, we have only established that POIE is occurs over ZnO, to a lesser extent over In2O3, and does not occur over SnO2. In the future, we may explore the long-term effect of the POIE and the effect of reductive annealing on it.

Reviewer 3 Report

A promising approach for the development of semiconductor gas sensors with reduced power consumption is their UV activation. In this work, the authors investigated the mechanisms of UV-activated processes on the surface of nanocrystalline ZnO, In2O3, and SnO2. The highest sensitivity is observed for ZnO. Mechanisms for this behavior are discussed with adequate investigation methods.

1. It might be beneficial to add around line 170 the surface topography of the oxides measured by AFM if available.

2. In fig. 2, 3, and 4 the ionic current for ZnO at 50 deg C is lower than the one at 30 deg C. Please comment.

3. Is Fig. 6 correct? From the figure, it seems that there is a change in the resistance after the O gas was removed!? If so - please explain. If not - shift the current graph to the left.

4. Rewording - line 306

Author Response

A promising approach for the development of semiconductor gas sensors with reduced power consumption is their UV activation. In this work, the authors investigated the mechanisms of UV-activated processes on the surface of nanocrystalline ZnO, In2O3, and SnO2. The highest sensitivity is observed for ZnO. Mechanisms for this behavior are discussed with adequate investigation methods.

Thank you for positive evaluating our article.

  1. It might be beneficial to add around line 170 the surface topography of the oxides measured by AFM if available.

Answer: Unfortunately, we do not have AFM, and the search for the AFM and operator will take a lot of time.

  1. In fig. 2, 3, and 4 the ionic current for ZnO at 50 deg C is lower than the one at 30 deg C. Please comment.

Answer: This is a good question; it can be seen that the POIE activity for ZnO behaves non-linearly - it decreases at 50°C, then increases at 100 and decreases again at 150°C. Apparently, there is competition between the processes of oxygen thermal desorption and participation in POIE processes. Probably, weak heating (up to 50°C) first leads to thermally activated oxygen desorption from the ZnO surface, as a result of which the oxygen concentration on the surface decreases and the rate of oxygen exchange decreases. Then, when heated to 100°C, oxygen photoadsorption again becomes energetically favorable, which leads to an increase in the exchange rate. The following phrase has been added to the text: “The dependence of the POIE rate on temperature for ZnO is clearly non-linear, decreasing upon heating to 50oC compared to room temperature, increasing upon heating to 100oC, and decreasing again at 150oC. Apparently, this behavior is associated with the dependence of the rate of thermally activated oxygen desorption from the surface of ZnO on the temperature.”

  1. Is Fig. 6 correct? From the figure, it seems that there is a change in the resistance after the O gas was removed!? If so - please explain. If not - shift the current graph to the left.

Answer: We have verified that the alignment of the impulses in the figure 6 below strictly corresponds to the peaks of the sensor response. In the presence of oxygen, the resistance of the sensors increases, then, in the absence of oxygen, it gradually decreases. The stepwise change in the resistance of the sensor during measurements (at 270, 480 and 690 min) is associated with increasing of sensor temperature, but not with oxygen in gas phase.

  1. Rewording - line 306

Answer: Thank you for your comment. The phrase

“UV irradiation leads to different effects on the sensor signal for three other metal oxides.”

 has been changed to

“UV irradiation affects the sensor signal differently for the various three metal oxides.”

Reviewer 4 Report

Well done, but ...

1) line 122: there is "aluminia" instead of "alumina", which means that the current would not flow from electrode to electrode through the nanocrystalline gas-sensitive layer, but through the carrier plate,

2) why such different samples: others for MS and others for POIE, I understand that there was a larger surface area for MS, but it is not known how these tablets in MS lay and what this surface really was, whether it was not twice as large, because from the illustration in [42] it may result (plus the lateral surface),

3) these are two completely different experiments (MS and POIE) with different samples, maybe if the same structures were used for MS as for POIEs,

4) hardly legible graphs, thicker lines should be and perhaps points, Fig. 6 and 7: very unfortunate dosing times and scale on the timeline, temperature axis also poorly chosen scale (ticks)
5) how to understand the reported energies of the metal-oxygen bond (lines 481-483) and the sequence from line 486?

6) the authors did not explain why they studied such metal oxides. While ZnO and SnO2 are well known, In2O3 is a rather low-prospective material (little indium and quite expensive). Gallium oxide Ga2O3 would probably be better...

Author Response

Well done, but ...

1) line 122: there is "aluminia" instead of "alumina", which means that the current would not flow from electrode to electrode through the nanocrystalline gas-sensitive layer, but through the carrier plate,

Answer: Thank you very much for your comment, you are absolutely right. A typo was made, "Aluminia" was replaced by "Alumina".

2) why such different samples: others for MS and others for POIE, I understand that there was a larger surface area for MS, but it is not known how these tablets in MS lay and what this surface really was, whether it was not twice as large, because from the illustration in [42] it may result (plus the lateral surface),

Answer: Thank you for your comment. In our work, we used two types of samples: thick films for oxygen sensor measurements and tablets for mass spectrometric measurements. Mass spectrometric measurements included the study of photoadsorption and photoactivated oxygen isotope exchange (POIE). The location of the tablets in the mass spectrometric cell corresponded to the drawings and the irradiated surface was accurately calculated (12.5 cm2)

Although it would be desirable to combine measurements of sample resistance and mass spectrometric measurements, this has not yet been realized. However, we can compare data because both films and tablets are made from the same material.

3) these are two completely different experiments (MS and POIE) with different samples, maybe if the same structures were used for MS as for POIEs,

Answer: The purpose of the work was precisely to compare the results of various experiments (sensor measurements to oxygen and photoadsorption/oxygen exchange). And we have shown that there is a correlation between the results of these experiments (see Figure 9). Although different samples were used (films and tablets), we can compare the results since the films and tablets were made from the same material.

4) hardly legible graphs, thicker lines should be and perhaps points, Fig. 6 and 7: very unfortunate dosing times and scale on the timeline, temperature axis also poorly chosen scale (ticks)

Answer: Figures 6 and 7 have been re-made according to your suggestions.

5) how to understand the reported energies of the metal-oxygen bond (lines 481-483) and the sequence from line 486?

Answer: The method for calculating the M-O energies is given in work [67] (there is a link in the text). In our work, we would not like to delve into this explanation and we would like to confine ourselves to providing a corresponding reference. Briefly, the M-O energy was calculated according to the formula: EM-O = {−ΔfH°(MOx) + ΔsubH°(M) + x/2•ΔdisH°(O2) + x•Δel.af.H°(O) + ΣΔion.H°(M)}/(x•CNO), where ΔsubH°(M)—sublimation enthalpy of metal, ΔdisH°(O2)—dissociation enthalpy of oxygen molecule, Δel.af.H°(O)—electron affinity of oxygen atom, ΣΔion.H°(M)—sum of ionization potentials of metal cation, CNO—coordination number of oxygen in the oxide. The M-O binding energies calculated by this formula are given by the sequence ZnO<In2O3<SnO2, i.e. from zinc oxide to tin oxide, the binding energy increases.

6) the authors did not explain why they studied such metal oxides. While ZnO and SnO2 are well known, In2O3 is a rather low-prospective material (little indium and quite expensive). Gallium oxide Ga2O3 would probably be better...

Answer: Indium oxide is one of the promising sensor materials because has a high electrical conductivity, although it is really expensive and rare. On the contrary, gallium oxide (its various crystalline modifications) has a high resistance, which makes it difficult to use it in low-temperature semiconductor gas sensors. If we focus on search queries in Scopus, then query "SnO2 AND gas AND sensor" gives 5385 records, query "ZnO AND gas AND sensor" gives 5534 records, query "In2O3 AND gas AND sensor" gives 1168 records, query "Ga2O3 AND gas AND sensor" gives only 203 records. Thus, we have relevantly selected the most used oxides for gas sensors.

Round 2

Reviewer 1 Report

This manuscript is suggested to publish in present form.

Author Response

Thank you for your positive evaluation of our article.